# Retirement Preparedness of Generation X Compared to Other Cohorts in the United States

Jia Qi, Swarn Chatterjee * and Yingyi Liu

Department of Financial Planning, Housing and Consumer Economics, University of Georgia, 205 Dawson Hall, Athens, GA 30602, USA; jiaqi@uga.edu (J.Q.); yyliu@uga.edu (Y.L.)
*   Correspondence: swarn@uga.edu

**Abstract:** According to the U.S. Census records, 40% of the population is aged between 35 and 64. This statistic means that a substantial percentage of the nation's population is in the wealth-formation phase of their life cycle and should be saving towards their retirement goals. Hence, the demand for retirement planning is anticipated to increase over the next decade. However, many economists and policymakers are concerned that a substantial number of American households are not well prepared for retirement. The Retirement Confidence Survey of the Employee Benefit Research Institute found that 36% of workers do not have any retirement savings. In particular, Generation X is the cohort that is least prepared for retirement. This research focuses on Generation X (40–54 years old) and explores this cohort's retirement preparedness relative to their Baby Boomer and Millennial peers. The study also models cohort effects and identifies the key factors affecting retirement preparedness. The result indicates that Generation X is better prepared for retirement than Millennials in safer portfolio allocations, but there is no significant difference in retirement adequacy between Gen Xers and Baby Boomers. Income, risk tolerance, and attainment of a college education are positively associated with retirement preparedness.

**Keywords:** retirement preparedness; cohort differences; cohort effects; portfolio returns

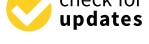



## 1. Introduction

Retirement planning is vital and necessary for almost every working individual. After retirement, individuals' income from work stops; however, they still have various expenses, including housing, grocery, medical, and other ordinary expenses in life. In this sense, people need to prepare in advance to fulfill their future financial needs. Retirement savings provide a source of financial security in retirement to help maintain peoples' pre-retirement lifestyle and well-being. This paper will focus on Generation X and investigate whether this cohort is prepared for retirement compared to Baby Boomers and Millennials. The demarcations of time range defining the generations are closely formulated by the literature: Baby Boomers were born between 1946 and 1964, Generation X were born between 1965 and 1980, Millennials were born between 1981 and 1996 (Cennamo and Gardner 2008; Twenge 2010; Dimock 2019). The reason to select Generation X is that workers in this cohort were found to be the least prepared for retirement according to a media report by Hill (2020). The Hill (2020) report mentioned that Generation X had higher levels of mortgage and personal debt obligations than other generational cohorts. Moreover, the Generation X cohort were the next generational cohort approaching retirement. In the 2019 Survey of Consumer Finances (SCF) dataset, which is maintained by the Federal Reserve, and is used in this study, the Generation X respondents were between 40–54 years old. In this sense, the results from this study should inform policy and be useful for explaining the retirement adequacy and preparedness of Generation X relative to the other generations.

This study examines the retirement preparedness of different generations and quantifies the generation effects. The model regresses the generational cohort indicators on

retirement preparedness while controlling demographics, socioeconomic factors, and psychological measures. This study identifies the key factors affecting retirement preparedness. The originality of this research is that it constructs unique measurements of retirement adequacy for different respondents, which means that each household will have a unique retirement baseline based on their current age, current income, expected retirement age, and remaining work-life expectancy (RWLE)[1].

Also, this research considers the purchasing power of money, which indicates that at different timelines, different individuals should have different baselines for retirement to maintain their previous consumption levels. For example, a 30-year-old individual with $50,000 in retirement savings is on the right track to retirement, but $50,000 is far less than enough for a 60-year-old consumer. It is more reasonable to construct unique baselines based on personal situations but not use the same standard for all consumers, which is often applied in the previous literature. Other than that, this study assumes different portfolio returns based on various risk-tolerance levels of different groups. For example, younger generations might want to take more risk in investment, so they will prefer to invest more of their savings into equity rather than fixed-income products. This strategy will adjust risk-tolerance levels to make the results more reliable.

The research will be an update of previous studies on the retirement saving behaviors of Generation X. Most of the earlier literature states that the retirement saving behavior of Generation X is unfavorable, and this generation is often less prepared for retirement. The result of the study is consistent with this statement but also has some disparities. This paper indicates that Generation X is better prepared for retirement than Millennials in safer asset allocations, but there is no significant difference between Generation X and Millennials. Furthermore, the result of the model shows that income and risk tolerance are positively related to retirement preparedness. Also, educational attainment will make a difference in affecting retirement preparedness.

This article provides a review of previous literature about retirement preparedness, and then introduces the theoretical framework of the model used in the research. Data, variable construction, hypotheses, and empirical models are discussed in the method section. Then, the results are presented and are followed by discussions in the last part of this article.

## 2. Literature Review

Findings from previous studies have implied that the demand for retirement planning will increase in the marketplace. One significant reason for the growing need for planning is that the defined benefit pension plans are being gradually replaced by defined contribution plans. Poterba (2014) finds that people's access to, and enrollments in, pension plans have steadily decreased over time. Pension plans or defined benefits plans are retirement vehicles that provide a guaranteed actuarially determined distribution to retirees. The distribution is computed using a formula based on the employees' number of years of service and the average of their highest 3–5 years of salaries, and the calculations may vary depending on the plan provider (Poterba et al. 2007). However, with pension plans being rapidly replaced by defined contribution-type retirement plans, the responsibility to save and generate savings for retirement has shifted to the employees (Butrica et al. 2009). In defined contribution plans, such as the 401K plans that are employer-sponsored retirement plans for American workers (Pence 2001), the employees must save and manage their investment portfolios within their 401K plan (Ippolito 1995). Therefore, to adequately prepare for their retirement, working adults should know how much they need for retirement and how much they need to save periodically, in order to meet their post-retirement consumption needs. Since the defined contribution plan is now more prevalent, working adults are responsible for acquiring investment knowledge, accepting the underlying investment risks, and generating adequate savings for their retirements.

However, the extant literature indicates that a substantial number of households are not adequately prepared for their retirement. According to the report of the Employee

Benefit Research Institute (2018), which is a nonpartisan, nonprofit research institute contributing to research on employee benefit programs and public policy, 36% of workers aged 25 and over did not have any savings for retirement. Also, according to the research of Munnell et al. (2018), almost half of all American working households expect to have inadequate retirement savings. This fraction rose from 31% in 1983 to 40% in 1998 and 50% in 2016. Similarly, in the retirement preparedness survey conducted by Prudential (2018), two in five respondents indicated that they did not know how much they would need monthly after retirement. The rapidly increasing proportion of under-prepared retirees could portend a sharp drop in retirees' purchasing power combined with a significant decrease in their financial well-being.

Cohort differences resulting from family and social backgrounds could impact financial well-being and retirement preparedness. College students' self-esteem was significantly improved between 1968 and 1994 (Twenge and Campbell 2001), which is within the time range of Generation X and Millennials. It is possible that consumers with higher self-esteem will be better prepared for retirement. The evidence from past literature indicates that personality traits are affected by generational effects. Of the Big Five personality factors, Extraversion, Agreeableness, and Conscientiousness are increasing with age, but Neuroticism tends to decrease (Smits et al. 2011). High expectations, materialism, and self-satisfaction also increase over generations (Twenge and Campbell 2010). These characteristics due to cohort effects could potentially influence consumers' financial well-being and retirement preparedness.

Compared with Baby Boomers, Generation Xers were less prepared for retirement based on most of the past literature. The study by the institution of Retirement Living (2021) found that only 35% of Gen Xers thought they would have enough savings for retirement compared to 75% of Baby Boomers. Although Gen Xers are younger and are expected to have more favorable views about investments, the Prudential (2018) study found that Generation X had less investment in retirement capitals than Baby Boomers. A study by Fidelity (2013) including Millennials found that Generation X was worse prepared for retirement than Baby Boomers but was better prepared than Millennials. Jackson and Hohman (2019) also found that the Baby Boomers were better prepared for retirement than Generation Xers and Millennials.

There is a concern that the lack of retirement preparedness among the younger generational cohorts may be detrimental to their well-being in retirement. This article uses a new approach to compute retirement adequacy and updates the findings from previous studies, using the most recent 2019 Survey of Consumer Finances (SCF) dataset. Additionally, the research also quantifies and controls for the cohort effects.

## 3. Theoretical Framework

Life-cycle theory (Ando and Modigliani 1963) and the theory of planned behavior (Ajzen 1985) are used to construct a framework for explaining the retirement preparedness of different generations. The life-cycle theory suggests that individuals are planning their consumption and savings and would even out their consumption over their lifetime (Ajzen 1985). The key assumption of this theory is that individuals tend to keep the same consumption level to maintain a stable lifestyle. This assumption provides theoretical background in constructing retirement preparedness in this paper, which assumes that individuals will keep the same consumption level before and after retirement, so that they would keep the same replacement ratio over their lifetime. This theory explains the behavior that individuals tend to save some part of their income while working and then spend the savings after retirement to keep their consumption level at the same over life.

The theory of planned behavior states that an individual's intention is shaped by three main components: attitudes (how a person thinks about a particular behavior), subjective norms (how most people think about a particular behavior), and perceived behavioral control (the person's perception about how difficult it is to perform the behavior) (Ajzen 1991). Based on this theory, generational differences in retirement savings should exist because

different generations have various attitudes, subjective norms, and perceived behavioral control. The behavioral intention in saving for retirement is hypothesized to be influenced by attitudes toward retirement savings, subjective norms, and perceived behavioral control, and the behavioral intention could explain the actual retirement savings behavior.

## 4. Materials and Methods

### 4.1. Data

This study uses the SCF 2019 dataset. The SCF dataset is maintained by the Federal Reserves and includes 22,615 observations with 4523 households (five implicates). The sample contains Millennials (age 22–36), Generation X (age 37–51), and Baby Boomers (age 52–67) under the full retirement age. The respondents of the survey were restricted to the primary income earners in the household. Retirement preparedness is measured by comparing the percentage of current retirement assets to the baseline to identify the percentage of respondents who have either achieved or those farther away from the estimated baseline. The retirement baseline is simulated using the computed present value of future retirement income based on several assumptions.

### 4.2. Variable Construction

#### 4.2.1. Wage Replacement Ratio (WPR)

The wage replacement ratio explains the percentage of income needed to maintain the same living standard upon retirement (Purcell 2012). The accepted rationale for the replacement ratio is between 70–85% (Vanguard Group 2019). According to the research of Finke et al. (2011), high-income earners will not have a high wage replacement ratio. High-income workers tend to have higher retirement savings, and they might not need that much savings to replace retirement. In contrast, since lower-income workers usually need a larger proportion of income for necessities, they would require a higher income replacement ratio than high-income earners (Purcell 2012). As most respondents in the SCF dataset are very wealthy with very high incomes, the WPR is assumed at 70% in this research.

#### 4.2.2. Retirement Age

According to the information from Social Security Administration (2021), the full retirement age is 67 for workers born after 1960. This article uses 67 as the retirement age because most respondents in the sample were born later than 1960.

#### 4.2.3. Inflation Rate

Since the current sample contains respondents who were 22 to 67 years old, the average inflation rate is assumed as an annualized rate in the past 40 years from 1979 to 2019. Based on the historical data about the inflation rate by U.S. Inflation Calculator (2021), the annualized inflation rate from 1979 to 2019 is 3.20%. The effect of purchasing power will be considered using this historical inflation ratio.

#### 4.2.4. Four-Percent Rule

The four-percent rule was first introduced by Bengen (1994), and it stands for the safe withdrawal rate from retirement portfolios when assuming that the minimum requirement of portfolio longevity is 30 years. Rule 25 was evolved from the four-percent rule in predicting the total retirement savings in the first year of retirement (Munnell et al. 2011; Thajudeen 2013). Given the four-percent rule calculates the annual withdrawal after retirement based on total savings, if the annual needs of post-retirement income were known, the amount needed in retirement would be identified.

#### 4.2.5. Portfolio Return

As consumers can invest their savings in various kinds of products with different rates of return, it is parsimonious to estimate the future value of retirement resources by

applying a single rate of return for all consumers (Montalto 2001). This research uses three scenarios in defining portfolio returns of retirement savings: 60/40 allocation (60% Equities; 40% Fixed Income), 70/30 allocation (70% Equities; 30% Fixed Income), and 80/20 allocation (80% Equities; 20% Fixed Income). The 60/40 portfolio would help increase expected returns while mitigating risks by diversifying investments (McQuinn et al. 2021), and this strategy can generate higher returns than stocks or bonds in the past 30 years. The study adds the other two allocation strategies to adjust for various risk-tolerance levels of different cohorts. For example, younger generations might prefer taking more risks and constructing riskier portfolios. The historical average return (PFR) was computed using the following formula:

$$\text{PFR}_i = w_i \text{Avg.(SPR)} + (1 - w_i)\text{Avg.(TR)} \tag{1}$$

$$i = 1,\ 2,\ 3,\ w_1 = 0.6,\ w_2 = 0.7,\ w_3 = 0.8$$

where,

Avg.(SPR) = Average return of the S&P 500 index from 1928–2016
Avg.(TR) = Average return of 10-year Treasuries from 1926–2016

Using the data provided by Damodaran (2022) from 1928 to 2016, the average return of the 60–40 portfolio is computed to be 7.78%, the average return of the 70–30 portfolio is 8.26%, and the average return of the 80–20 portfolio is computed to be 8.74%.

### 4.2.6. Social Security

Since social security typically replaces approximately 40% of pre-retirement income (Biggs and Springstead 2008; CBPP 2022), this research assumes that at the full retirement age of 67, 40% of the total future retirement savings will be social security.

### 4.2.7. Dependent Variable

This research's dependent variable is retirement preparedness, which is defined as the ratio of current retirement savings over the baseline of each respondent's unique baseline of savings. Firstly, based on the above assumptions, the expected annual retirement income needed (EARN) to maintain previous living standards will be computed as:

$$\text{EARN}_j = \left(\text{Income}_j * \text{WPR}\right) * \left[(1 + \text{Inflation})^{(67 - \text{current age})_j}\right] \tag{2}$$

After that, the baseline will be identified by discounting the future cash flows to the current age while considering the return of retirement portfolios and the impact of purchasing power.

$$\text{Inflation Adjusted Return Rate}_i = \left(\frac{1 + \text{PFR}_i}{1 + \text{inflation}} - 1\right) \tag{3}$$

$$\text{Baseline}_{ij} = \frac{\text{EARN}_j * 25}{[1 + \text{IARR}_i]^{\text{RWLE}_j}} \tag{4}$$

$$\text{Current Retirement Assets}_{ij} = \frac{\left\{\text{FinAssets}_j * [1 + \text{Adjusted } R_i]^{\text{RWLE}_j}\right\}/\text{Social Replace}}{(1 + \text{Inflation})^{\text{RWLE}_j}} \tag{5}$$

where

Remaining Work Life Expectancy (RWLE) = 67 − Current age
$i$ is the indicator of portfolio strategy, $i$ = 1, 2, 3
$j$ is the respondent, $j$ = 1, 2, . . . , $n$

In this sense, each household will have a unique baseline of retirement savings. Then the retirement preparedness will be measured as:

$$\text{Retirement Preparedness}_{ij} = \frac{\text{Current Retirement Assets}_{ij}}{\text{Baseline}_{ij}} * 100\% \tag{6}$$

Based on this measurement, a larger value of the retirement preparedness means that the respondent is better prepared.

### 4.2.8. Independent Variables of Interest

The independent variables of interest in this study were the generational cohort-related variables including Millennials, Baby Boomers, and the Generation X. The cohort indicators were dummy variables that were coded as 1 = YES; 0 = NO.

### 4.2.9. Other Control Variables

The other independent control variables comprised of demographics, socioeconomic factors, and psychological measures. The control variables include age, gender, race, marital status, income, household size, remaining work-life expectancy, education level, health conditions, risk tolerance, financial literacy, and remaining life expectancy. Table A1 shows the coding and description of all the control variables.

### 4.3. Hypotheses

**H1.** *Retirement preparation of the households will vary by generational cohort after controlling for other socioeconomic, demographic, and income related characteristics.*

**H2.** *Retirement preparation of the households will vary by generational cohort, across various asset allocation scenarios, after controlling for other socioeconomic, demographic, and income related characteristics.*

### 4.4. Empirical Models

This study first visualizes the means or frequencies of some key factors across cohorts. For example, how retirement preparedness, baseline, and retirement assets are different for each generation, and whether the respondents meet their retirement needs. If the value of retirement preparedness is larger than 1, the respondent has reached the retirement baseline. Then, the study conducts an ANOVA analysis to get the distribution of retirement preparedness among cohorts so that the cohort differences could be identified.

Since retirement preparedness is created as a continuous variable, the research uses Ordinary Least Square (OLS) regression to determine relationships. The formula of the OLS regression is:

$$f = \beta_0 + \beta_1 C + \beta_2 X + \varepsilon \tag{7}$$

where $f$ is retirement preparedness, $C$ is cohort indicator that is the vector of three binary variables, $\beta_i X$ is the vector denoting control variables, and $\varepsilon$ is the error term.

The analyses for this study using the SCF 2019 dataset was conducted adjusting for the five implicates to impute for missing data using multiple imputation techniques and replicate weights (Board of Governors of the Federal Reserve System 2020). Multicollinearity has been tested after the regression, and the results indicate that the VIFs of the variables included in the model was under 3.0, which was not a concern for multi-collinearity (Wooldridge 2015).

## 5. Results

### 5.1. Descriptive Statistics

The descriptive statistics are presented in Table A2. The results indicate that the average age of the participants was approximately 44 years (43.881). The ages ranged from 22 to 67 years. The average household income was $106,428.50. The average household

size was 2.77, 26% of the respondents were female, and 56% were married. On average, the respondents got 2.168 of the Big 3 financial literacy questions correct. The average risk tolerance score was 5.014. The results from Table A3 suggest that the later cohorts have higher mean baselines, retirement assets, and better preparedness. Also, the percentage of having already met the retirement baseline increased with age. However, older generations have lower mean education levels and worse health conditions.

### 5.2. ANOVA Model

The one-way ANOVA identifies significant cohort differences in retirement preparedness. Table A4 illustrates the result of ANOVA analysis. Based on the ANOVA results, the hypothesis that retirement preparedness is the same by cohorts is rejected. The results from Tukey's test revealed that there are significant differences in retirement preparedness among all groups.

### 5.3. OLS Model

The OLS regression results for retirement preparedness are presented in Tables A5–A7. The results in Table A5 (60–40 allocation) indicate that compared to the reference group of the Generation X cohort, the respondents in the Millennial cohort were less likely (beta = $-0.045$; SE = 0.020) to be prepared for retirement. Income (beta = 0.035; SE = 0.011) and risk tolerance (beta = 0.011; SE = 0.005) were also positively associated with retirement preparedness. Compared to the reference group of respondents with the educational attainment of high school or lower, those with college degrees were more likely to be prepared for their retirement (beta = 0.089; SE = 0.019).

Results in Table A6 show the associations between retirement preparedness with a 70/30 equity and fixed income allocation. The results indicate that relative to the Generation-X cohort, the Millennials were negatively associated with preparedness even with a 70/30 allocation. Income (beta = 0.038; SE = 0.012), risk tolerance (beta = 0.011; SE = 0.004), and attainment of college education (beta = 0.103; SE = 0.021) were also positively associated with retirement preparedness.

Results in Table A7 show the associations between retirement preparation if 80/20 equity and fixed income allocation was followed. The results indicate that income (beta = 0.421; SE = 0.012), risk tolerance (beta = 0.012; SE = 0.005), Financial literacy (beta = 0.008; SE = 0.005) and attainment of college education (beta = 0.118; SE = 0.000) are positively associated with retirement preparation. However, there is no significant difference between cohorts at this higher equity allocation.

### 5.4. Limitations

There were several limitations to this study. This study assumed that individuals kept constant consumption patterns, however, there is evidence in literature that retirees reduced consumption and expenditure due to home production, and other constraints related to time to search for savings (Lührmann 2010). Caution should also be applied when generalizing the results of this study since approximately 26% of the sample in the SCF dataset were women. In the SCF dataset, the wealth-related data was available at household level, but the other sociodemographic variables were available at individual level. Although the information was restricted to the primary income earners in this study, this is a potential limitation of using the SCF dataset.

## 6. Discussions and Conclusions

The significant findings of this study add to the literature on differences in retirement preparedness by generational cohorts. Applying the empirical model based on the life-cycle theory and the theory of planned behavior, this research confirms the results from previous studies that cohort effects were significantly associated with differences in retirement preparedness (Jackson and Hohman 2019). In particular, after controlling for various socioeconomic, demographic, and behavioral factors, our results suggest that Generation

X maybe better prepared for retirement than the Millennials (Jackson and Hohman 2019) in general asset-allocation situations. Additionally, income, risk tolerance, and college education attainment are positively associated with retirement preparedness. Our findings add to the literature on retirement planning and contribute to the broader discussion on the generational differences in retirement preparedness, as was discussed in the Hill (2020) report.

This research utilized a new mathematical approach for computing retirement adequacy. Instead of introducing the same standard of retirement preparedness for all the households as done in past literature, the unique baseline of different households works well with the dataset, and most of the results are in line with the expectation. It is more reasonable to compare individuals' current retirement assets to their own standard based on their current age, income, expected retirement age, remaining work-life expectancy (RWLE), remaining life expectancy (RLE), and asset allocation strategies.

While most of the past literature has measured consumers' retirement adequacy using the same portfolio return rate (Kim et al. 2014), this research introduces the asset allocation strategy when examining individuals' retirement preparedness, as consumers might have varying investment decisions and priorities in different stages of their life cycles. This strategy adjusts for risk-tolerance levels of different consumers as an additional control for behavioral financial differences in the model. Our study shows that Generation X is better prepared for retirement than Millennials in widely used 60/40 and 70/30 strategies. However, when consumers are assumed to take risks by investing more in equity, there is no difference in retirement preparedness across cohorts. When consumers anticipate earning a high rate of return for their retirement savings, their baseline of retirement preparedness is lower because the savings are assumed to grow faster. In this sense, consumers might believe they have already saved enough because of the high return rate, so the differences in retirement preparedness among different cohorts might not be significant. However, consumers should understand that more investment risks will come with high expected returns. Financial planners should recommend that clients set up enough emergency funds when choosing high-risk portfolios to offset the risks in investments. On the other hand, over-allocating one's portfolio into lower-risk assets generally comprises one's emergency fund reserves, which comes with high opportunity costs because clients cannot invest these funds in any other retirement accounts with higher return rates. So, financial planners need to comprehensively evaluate clients' situations when they invest in high-return portfolios, such as clients' monthly living expenses, income stability, and health situations.

The findings from the results of this study infer that consumers might understand the importance of retirement savings as they become older, and retirement preparations and savings likely gain more priority and salience for people as they progress with age. Consumers may not worry about retirement savings when they are younger, but by the time they realize the importance of saving for retirement, they are likely to find themselves constrained by their investment time horizon, as well as by their risk capacity and tolerance, to achieve their retirement goals. Therefore, the findings from this study confirm that people should be encouraged to begin the process of long-term savings earlier in their lives (Bongini and Cucinelli 2019). Starting early on saving for one's retirement, and saving on a regular basis, provides the best opportunity for individuals to accumulate their retirement wealth over time (Farhi and Panageas 2007). Besides, beginning the investment process early can help individuals mitigate their risk of inadequate savings as people approach retirement. When compared with people who either do not save for their retirement or begin the process of saving very late in their working lives, individuals who begin saving early may feel less pressure associated with needing to add riskier asset classes to their portfolios later on in their lives, which may or may not be compatible with their risk tolerance, as they approach retirement. Additionally, our findings indicate that increasing human capital, including educational attainment and financial literacy, could help consumers prepare earlier and better for retirement.

Future studies need to apply the methodology for calculating retirement preparedness from this study for estimating retirement preparedness using other datasets. Moreover, this study was restricted to only the U.S. generational cohorts, but this research can be broadened, and similar research is needed for examining the retirement preparedness of individuals in other parts of the world, especially in Europe and Asia, where a large segment of the population is either retiring or is rapidly approaching retirement. This study specifically focused on generational cohort-related differences, but future studies need to examine whether other behavioral factors such as self-regulation, personality traits, or perceived financial capability also play mediating roles in the association between generational cohort-related differences and retirement preparedness of households.

As the demand for retirement-planning services increases, the results of this study have implications for financial planning professionals. This research could help promote the understanding of retirement saving behavior among different cohorts. Financial planners may provide varied advice based on the characteristics of cohorts when working with clients. For example, financial planners need to pay more attention to clients' retirement saving behaviors when working with Millennials, since the Millennials in this study were associated with being less prepared for retirement than Generation X. Advisors might always need to evaluate clients' current positions for retirement and encourage clients to start saving early and regularly. The findings also suggest that along with income, health conditions, and risk tolerance, educating clients on their financial situations should also be considered when making financial planning recommendations.

**Author Contributions:** Conceptualization, J.Q. and S.C.; Methodology, J.Q. and S.C.; Software, J.Q.; Validation, J.Q. and S.C.; Formal Analysis, J.Q. and S.C.; Investigation, J.Q. and S.C.; Resources, J.Q. and S.C.; Data Curation, J.Q.; Writing—Original Draft Preparation, J.Q., S.C. and Y.L.; Writing—Review and Editing, J.Q., S.C. and Y.L.; Visualization, J.Q. and S.C.; Supervision, J.Q. and S.C. All authors have read and agreed to the published version of the manuscript.

**Funding:** This research received no external funding.

**Institutional Review Board Statement:** Not applicable.

**Informed Consent Statement:** Not applicable.

**Data Availability Statement:** Raw data were generated at 2019 Survey of Consumer Finances (SCF). Derived data supporting the findings of this study are available from the corresponding author S.C. on request.

**Conflicts of Interest:** The authors declare no conflict of interest.

## Appendix A

**Table A1.** Description of the Variables used in the Study.

| Variable Name | Description | Type |
| --- | --- | --- |
| Age | 22 to 67 | Continuous |
| Generational Cohorts (Ref: Generation X) | | |
|    Baby Boomers | 1 = Yes; 0 = No | Binary |
|    Generation X | 1 = Yes; 0 = No | Binary |
|    Millennials | 1 = Yes; 0 = No | Binary |
| Household Size | 1 to 12 | Continuous |
| Female | 1 = Yes; 0 = No | Binary |
| Marital Status | | |
|    Married | 1 = Yes; 0 = No | Binary |
| Remaining Life Expectancy (RLE) | 0 to 121 | Continuous |
| Remaining Work Life Expectancy (RWLE) | 0 to 45 | Continuous |
| Income | 1018 to 676,000,000 | Continuous |
| Financial Literacy (Big 3) [1] | 0 = None correct to 3 = All Correct | Continuous |
| Risk Tolerance | 0 = Low to 10 = High | Continuous |

**Table A1.** *Cont.*

| Variable Name | Description | Type |
|---|---|---|
| Race/Ethnicity (Ref: Other race) | | |
|   White | 1 = Yes; 0 = No | Binary |
|   Black | 1 = Yes; 0 = No | Binary |
|   Hispanic | 1 = Yes; 0 = No | Binary |
|   Other race | 1 = Yes; 0 = No | Binary |
| Edu Attain. (Ref: High Schl or less) | | |
|   College | 1 = Yes; 0 = No | Binary |
|   Some Col | 1 = Yes; 0 = No | Binary |
|   High Schl or less | 1 = Yes; 0 = No | Binary |
| Health (Ref: Poor) | | |
|   Excellent | 1 = Yes; 0 = No | Binary |
|   Good | 1 = Yes; 0 = No | Binary |
|   Fair | 1 = Yes; 0 = No | Binary |
|   Poor | 1 = Yes; 0 = No | Binary |

[1] The "Big Three" Financial Literacy Questions (Lusardi and Mitchell 2014). (1) Suppose you had $100 in a savings account and the interest rate was 2% per year. After 5 years, how much do you think you would have in the account if you left the money to grow?

More than $102
Exactly $102
Less than $102
Do not know
Refused

(2) Imagine that the interest rate on your savings account was 1% per year and inflation was 2% per year. After 1 year, how much would you be able to buy with the money in this account?

More than today
Exactly the same as today
Less than today
Do not know
Refused

(3) Please tell me whether this statement is true or false. "Buying a single company's stock usually provides a safer return than a stock mutual fund."

True
False
Do not know
Refused

**Table A2.** Descriptive Statistics.

| Variables | Mean | Std. Dev. | Min | Max |
|---|---|---|---|---|
| Age | 43.881 | 12.192 | 22 | 67 |
| Generational Cohorts (Ref: Generation X) | | | | |
|   Baby Boomers | 39.31% | | 0 | 1 |
|   Generation X | 36.34% | | 0 | 1 |
|   Millennials | 24.35% | | 0 | 1 |
| HHSize | 2.772 | 1.474 | 1 | 12 |
| Female | 26% | | 0 | 1 |
| Married | 56% | | 0 | 1 |
| Remaining Life Expectancy (RLE) | 38.089 | 15.623 | 0 | 121 |
| Remaining Work Life Expectancy (RWLE) | 20.459 | 12.009 | 0 | 45 |
| Income | $106,428.50 | $455,399.500 | $1018 | $676,000,000.00 |
| Fin_Lit | 2.168 | 0.859 | 0 | 3 |
| Risk tolerance | 5.01 | 2.219 | 1 | 10 |
| Race/Ethnicity (Ref: Other race) | | | | |
|   White | 68.75% | | 0 | 1 |
|   Black | 13.37% | | 0 | 1 |
|   Hispanic | 11.40% | | 0 | 1 |
|   Other race | 6.47% | | 0 | 1 |

**Table A2.** *Cont.*

| Variables | Mean | Std. Dev. | Min | Max |
|---|---|---|---|---|
| Edu Attain. (Ref: High Schl or less) | | | | |
|   College | 47.99% | | 0 | 1 |
|   Some Col | 24.32% | | 0 | 1 |
|   High Schl or less | 27.69% | | 0 | 1 |
| Health (Ref: Poor) | | | | |
|   Excellent | 30.28% | | 0 | 1 |
|   Good | 51.50% | | 0 | 1 |
|   Fair | 16.66% | | 0 | 1 |
|   Poor | 1.56% | | 0 | 1 |
| N = 3709 | | | | |

**Table A3.** Descriptive Comparison by Cohort.

| | Baby Boomers (N = 1458) | | Gen X (N = 1348) | | Millennials (N = 903) | |
|---|---|---|---|---|---|---|
| Variable | Mean | Std. Dev. | Mean | Std. Dev. | Mean | Std. Dev. |
| Preparedness | 0.349 | 0.666 | 0.294 | 0.562 | 0.205 | 0.341 |
| Baseline | 2,407,838.00 | 13,300,000 | 1,541,343.00 | 4,043,114.00 | 628,914.20 | 880,549.30 |
| Educ | 2.975 | 0.981 | 3.005 | 1.003 | 3.011 | 0.961 |
| Fin_Lit | 2.346 | 0.798 | 2.204 | 0.860 | 2.043 | 0.878 |
| Health | 2.007 | 0.699 | 1.969 | 0.713 | 1.877 | 0.724 |
| Meet | 0.076 | 0.265 | 0.054 | 0.225 | 0.025 | 0.157 |

**Table A4.** ANOVA of Retirement Preparedness by Generational Cohorts.

| | Preparedness | | | |
|---|---|---|---|---|
| Cohort | Mean | St. Dev | Frequency | |
| BabyBoomers | 0.349 | 0.666 | 1458 | |
| GenerationX | 0.294 | 0.562 | 1348 | |
| Millennials | 0.205 | 0.341 | 903 | |
| | 0.282 | 0.541 | 3709 | |

**Between Group Variance**
*F-Stat: 21.91; p < 0.000*
**Within Group variance (Bartlett)**
*Chi(2) = 169.355; p < 0.000*

**Tukey Pairwise Comparisons of Means with Equal Variances**

| | Contrast | St. Err | Tukey | *p*-Value |
|---|---|---|---|---|
| Gen X vs. BB | −0.121 | 0.016 | −7.68 | <0.000 |
| Mill vs. BB | −0.331 | 0.018 | −18.75 | <0.000 |
| Mill vs. Gen X | −0.210 | 0.018 | −11.72 | <0.000 |

**Table A5.** Retirement Preparedness Estimation for Simulated 60–40 Allocation.

| Variables | Coef. | Std. Err. | Sig |
|---|---|---|---|
| Millennials | −0.045 | 0.020 | ** |
| BabyBoomers | 0.057 | 0.036 | |
| logIncome | 0.035 | 0.011 | *** |
| HHSize | −0.006 | 0.007 | |
| RLE | 0.000 | 0.001 | |
| RiskTol | 0.010 | 0.005 | ** |
| Fin_Lit | 0.006 | 0.004 | |
| Married | 0.025 | 0.027 | |

**Table A5.** *Cont.*

| Variables | Coef. | Std. Err. | Sig |
|---|---|---|---|
| Female | −0.021 | 0.032 | |
| White | 0.013 | 0.041 | |
| Black | −0.055 | 0.043 | |
| Hispanic | −0.043 | 0.045 | |
| College | 0.090 | 0.020 | *** |
| SomeCollege | 0.006 | 0.019 | |
| Excellent | 0.076 | 0.061 | |
| Good | 0.058 | 0.060 | |
| Fair | 0.032 | 0.058 | |
| Intercept | −0.432 | 0.127 | *** |
| R-squared = 0.2139 | | | |
| N = 3442 | | | |

*** $p < 0.01$; ** $p < 0.05$.

**Table A6.** Retirement Preparedness Estimation for Simulated 70–30 Allocation.

| Variables | Coef. | Std. Err. | Sig |
|---|---|---|---|
| Millennials | −0.042 | 0.022 | * |
| BabyBoomers | 0.043 | 0.038 | |
| Log Income | 0.038 | 0.012 | *** |
| HHSize | −0.007 | 0.008 | |
| RLE | 0.000 | 0.001 | |
| Risktol | 0.011 | 0.005 | ** |
| Fin_Lit | 0.007 | 0.004 | |
| Married | 0.031 | 0.029 | |
| Female | −0.027 | 0.034 | |
| White | 0.011 | 0.044 | |
| Black | −0.066 | 0.047 | |
| Hispanic | −0.054 | 0.049 | |
| College | 0.103 | 0.021 | *** |
| SomeCollege | 0.007 | 0.020 | |
| Excellent | 0.086 | 0.066 | |
| Good | 0.066 | 0.065 | |
| Fair | 0.037 | 0.063 | |
| Intercept | −0.478 | 0.135 | *** |
| R-Squared = 0.2031 | | | |
| N = 3442 | | | |

*** $p < 0.01$; ** $p < 0.05$; * $p < 0.10$.

**Table A7.** Retirement Preparedness Estimation for Simulated 80–20 Allocation.

| Variables | Coef. | Std. Err. | Sig |
|---|---|---|---|
| Millennials | −0.035 | 0.026 | |
| BabyBoomers | 0.026 | 0.040 | |
| logIncome | 0.042 | 0.012 | *** |
| HHSize | −0.007 | 0.009 | |
| RLE | 0.000 | 0.001 | |
| risktol | 0.012 | 0.005 | * |
| Fin_Lit | 0.008 | 0.005 | * |
| married | 0.038 | 0.031 | |
| Female | −0.034 | 0.036 | |
| White | 0.009 | 0.047 | |
| Black | −0.080 | 0.051 | |
| Hispanic | −0.067 | 0.053 | |
| College | 0.119 | 0.023 | *** |
| SomeCollege | 0.008 | 0.022 | |

**Table A7.** *Cont.*

| Variables | Coef. | Std. Err. | Sig |
|---|---|---|---|
| Excellent | 0.098 | 0.071 | |
| Good | 0.075 | 0.070 | |
| Fair | 0.044 | 0.068 | |
| _cons | −0.528 | 0.145 | *** |

R-Squared = 0.1945
N = 3442

*** $p < 0.01$; * $p < 0.10$.

## Note

[1] The remaining work-life expectancy (RWLE) is constructed using 67 (full-retirement age) minus the respondents' current ages. The remaining life expectancy (RLE) used in this study is constructed by subtracting the respondents' current ages from their self-reported life expectancy in years.

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
