# Peer review of "Retirement Preparedness of Generation X Compared to Other Cohorts in the United States"

_ijfs, doi:10.3390/ijfs10020045_

Round 1

Reviewer 1 Report

The topic of retirement preparedness and long-term savings is vital and becomes urgent with the aging of societies. The research presented in the article is interesting and bring a new insight to the problem. However, is not free from flaws. 

1. In the abstract and the introduction Authors use "our" as if readers knew what is the population they refer to. The fact that the research concerns the American pension system and this population should be better introduced. Also due to the international scope of the journal, the reader might not be familiar with the American saving for retirement financial instruments. It is recommended to explain the issues marked in the attached file in yellow.

2. Also, the generations' definitions need to be introduced. A sentence explaining the time range defining generations with valid sources should be sufficient. Please note, that different nations may slightly differ in classification into generations.

3. Similar comment for "SCF data". When the shortcut is first used it should be introduced. It should be explained that it is American Census.

4. The introduction lacks the organization of the article.

5. The style of numerous sentences brings confusion, often the context is unclear. The phrases and sentences requiring corrections or that should be considered for rephrasing were marked in green in the file.

6. The line of reasoning behind generational differences and its expected impact on retirement preparedness is poorly developed in last paragraph on p. 2. The literature support in this aspect is basic.

7. In first paragraph on p. 3 some findings are repeated. There is no need for repeating the same finding twice - for each source separately. You should group the sources supporting the same finding, which facilitates the reader understanding the discussion.  

8. The word cohort is used as synonim to the generation. It is not justified enough in the introduction. In the last paragraph of section 3 Authors should consider using notion 'generation' instead of 'cohort' while discussing theoretical background. Also, it is recommended to use text navigation to support line of reasoning (marked in red in the file). 

9.  Formulas (1)-(3) is not one formula. It would be good to add subscript like 60/40 by PFR in first and so on. Alternatively it could be one equation  with subscript "i" denoting the number indicating the strategy.

In new line "Where" should not be capitalized no indent in this paragraph and no comma after the word. Instead of  "SPR=Average..." it should probably be "Avg. (SPR) - average return... 2016, Avg.(TR) - average return... 2016." (is it TR or TBR?) 
or "Avg. (SPR) is the average return... 2016, Avg. (TR) it the average ... 2016." Please pay attention to the punctuation and capital letters in mathematical notation. 

  10. In the dependent variable explanation the subscript denoting the subject is missing, eg. j denotes the j-th respondent and then EARN_j etc. 

  11. In equation 5 the "portfolio return" is the same as PFR in equations (1-3)?

  12. Shortcut RWLE should be explained by the equation 6. Is this the same as (67-current age) or is this some random variable?

  13. Is "Adjusted R" (equation 6) the same as Inflation Adjusted Return Rate (equation 5)? Is this fixed for all respondents, or respondent-variable? What is "Social Replace" (equation 7)?

  14. "Current retirement Assets" is the same as "Current Retirement"? (equations 7 and 8).

15. There is hypothesis about the fisk tolerance and financial literacy but the notions and its measures are not addressed in the literature review and the discussion. Also, using the conjunction "and" between them in the hypothesis is rather the misuse. 

16. The description under the equation 9 could be better developed. C is probably the vector of 3 binary variables indicating cohorts, then /beta_1 is also the vector. X is the vector of control variables.  

17. The terms used for naming generations should be unified. Should Millennials be capitalized or not, etc.?

Reading the article one might wonder, how it is possible to separate the effect of generation from the effect of age. Fortunately, Authors discuss this aspect in the Conclusions, addressing the doubts. 

Author Response

Dear Editor,

We thank you for providing us the opportunity to revise and resubmit our manuscript. We also want to thank both the reviewers for their valuable suggestions, incorporating which have helped in substantially improving our manuscript. We have been able to address all of the reviewers’ concerns. Our point-to-point responses to the reviewer suggestions and comments are listed below.

Kind Regards,

Jia Qi

Swarn Chatterjee (Corresponding Author)

Yingyi Liu

Reviewer 1

  1. In the abstract and the introduction Authors use "our" as if readers knew what is the population they refer to. The fact that the research concerns the American pension system and this population should be better introduced. Also due to the international scope of the journal, the reader might not be familiar with the American saving for retirement financial instruments. It is recommended to explain the issues marked in the attached file in yellow.

[Response] This is a great advice. We have made several changes to make it clear for readers to understand the scope.

  1. Also, the generations' definitions need to be introduced. A sentence explaining the time range defining generations with valid sources should be sufficient. Please note, that different nations may slightly differ in classification into generations.

[Response] Time ranges are added with sources.

  1. Similar comment for "SCF data". When the shortcut is first used it should be introduced. It should be explained that it is American Census.

[Response] The SCF dataset is maintained by the U.S. Federal Reserve. We have made this change in the manuscript.

  1. The introduction lacks the organization of the article.

[Response] Organization has been added to the introduction. We thank the reviewer for this suggestion.

  1. The style of numerous sentences brings confusion, often the context is unclear. The phrases and sentences requiring corrections or that should be considered for rephrasing were marked in green in the file.

[Response] All of the marked sentences have been adjusted accordingly.

  1. The line of reasoning behind generational differences and its expected impact on retirement preparedness is poorly developed in last paragraph on p. 2. The literature support in this aspect is basic.

[Response] Adjustments have been made but, the purpose of this paragraph is to introduce the family and social backgrounds that can potentially impact retirement preparedness over generations. Also, it helps to conceptually justify our model in constructing unique baselines for each respondent but not use the same standard.  We are not talking about the exact or expected impact from individual characteristics because they are not the aim of this research. We are focusing on the cohort differences.

  1. In first paragraph on p. 3 some findings are repeated. There is no need for repeating the same finding twice - for each source separately. You should group the sources supporting the same finding, which facilitates the reader understanding the discussion.  

[Response] Adjustments have been made.

  1. The word cohort is used as synonim to the generation. It is not justified enough in the introduction. In the last paragraph of section 3 Authors should consider using notion 'generation' instead of 'cohort' while discussing theoretical background. Also, it is recommended to use text navigation to support line of reasoning (marked in red in the file). 

[Response] Adjustments have been made.

  1.  Formulas (1)-(3) is not one formula. It would be good to add subscript like 60/40 by PFR in first and so on. Alternatively it could be one equation with subscript "i" denoting the number indicating the strategy.

In new line "Where" should not be capitalized no indent in this paragraph and no comma after the word. Instead of  "SPR=Average..." it should probably be "Avg. (SPR) - average return... 2016, Avg.(TR) - average return... 2016." (is it TR or TBR?) 
or "Avg. (SPR) is the average return... 2016, Avg. (TR) it the average ... 2016." Please pay attention to the punctuation and capital letters in mathematical notation. 

[Response] Adjustments have been made. We added the subscript “i” to indicate the portfolio strategy.

We have adjusted statements after “where” to Avg.(SPR) and Avg.(TR). It should be TR but not TBR, we are sorry that it is a typo.

  1. In the dependent variable explanation the subscript denoting the subject is missing, eg. j denotes the j-th respondent and then EARN_j etc. 

[Response] Subscripts have been added.

  1. In equation 5 the "portfolio return" is the same as PFR in equations (1-3)?

[Response] Yes, we have changed "portfolio return" to PFR.

  1. Shortcut RWLE should be explained by the equation 6. Is this the same as (67-current age) or is this some random variable?

[Response] RWLE = 67 - current age. Adjustments have been made.

  1. Is "Adjusted R" (equation 6) the same as Inflation Adjusted Return Rate (equation 5)? Is this fixed for all respondents, or respondent-variable? What is "Social Replace" (equation 7)?

[Response] Yes, Adjusted R is the same as Inflation Adjusted Return Rate. Its value depends on the selection of portfolio strategy, so we added subscript i to it.

Social Replace is the social replacement ratio that is fixed at 40% in this study. Please read 4.2.6. for details.

  1. "Current retirement Assets" is the same as "Current Retirement"? (equations 7 and 8).

[Response] Yes, we have changed the "Current Retirement" to "Current retirement Assets" to keep them same.

  1. There is hypothesis about the fisk tolerance and financial literacy but the notions and its measures are not addressed in the literature review and the discussion. Also, using the conjunction "and" between them in the hypothesis is rather the misuse. 

[Response] We have removed H03 because risk tolerance and financial literacy are not the focus of this research.

  1. The description under the equation 9 could be better developed. C is probably the vector of 3 binary variables indicating cohorts, then /beta_1 is also the vector. X is the vector of control variables.  

[Response] Adjustments have been made.

  1. The terms used for naming generations should be unified. Should Millennials be capitalized or not, etc.?

[Response] Following the reviewer’s suggestion, we have made these generational names consistently with capitalized letters at the beginning of each generational cohort’s name.

Reviewer 2 Report

Literature

  1. First line of abstract should state it is US data.
  2. The EBRI data is for what age group?
  3. 3. Briefly explain 401K in a footnote for non-American readers.
  4. 5. Social security statistics from 2008 – is this still currently applicable? I would suggest that social security support has changed since 2008.
  5. There is little/nothing in the way of literature to support any of your highlighted relationships in H03 between financial knowledge/literacy, education, risk tolerance and retirement preparedness (e.g. Wang & Bartholomae, 2020; Noviarini et al, 2021). This needs to be added.
  6. The authors explain what is new in their study. Thank you. Can you also differentiate it from Hill (2020)?
  7. Although you use Life cycle theory and assume that individuals will keep constant consumption patterns, it should be acknowledged that this is not always true. There is evidence that retirees reduce consumption and expenditure due to home production, time to search for savings etc (Lührmann, 2010).

Method and Results

There are a number of omissions that I would expect to see:

  1. Description/Table of how variables are measured (e.g. health, risk tolerance, Financial literacy (needs explanation and reference to “the Big 3”), financial knowledge (what is the difference between financial literacy and financial knowledge?), education level
  2. What are implicates?…after much searching I found the description in https://www.federalreserve.gov/econres/scfindex.htm. This should be explained in the paper.
  3. Independent variables (p.5; s 4.2.8)- it is unclear what are the independent variables and what are the control variables. It appears that cohorts are the independent variables for H01 and H02 and that risk tolerance and financial literacy are the independent variables for H03. However, the authors state that the independent variables include cohort indicators, demographics, socioeconomic factors, and phycological measures and risk tolerance appears in the list of control variables.
  4. 5. Eqn 8 What is “social replace”?
  5. 9. Table 2, what is “Meet”?

Other issues

  1. In the calculation of dependent variable “retirement preparedness”…in Eqn6, what is the significance of “25”?; what is “R”?
  2. SPR – is this calculated by the authors from 1926 (pp 4-5) or did you use Damodaran (2021) (p. 5) from 1929? Also why go back such a long time, when inflation is measured from 1979 onwards?
  3. 5. - how is “remaining life expectancy” measured? Is it adjusted for gender and race for each individual?
  4. H03- why has financial literacy just appeared? It was not mentioned in the literature section.
  5. It seems that the calculations are per household (p.1.near bottom of page; p.5. just above Eqn 8) but then some data is by individual (e.g. Eqn 4, 67-age; age variable, marital status, gender, health, education). How did you deal with these different measurement units? How did you combine individual data?
  6. You say you test for m How was this done and what are the results? For example, are age and cohorts highly correlated?
  7. In the testing of H03, there is the potential for endogeneity (reverse causality) (Goyal and Kumar (2021)) between financial literacy and financial outcomes (retirement preparedness). This should be considered.

Conclusions

  1. There is a bias in your sample – it is only quarter female (when population % would be around 50%) and it is high income (p.4). We know that women (particularly single women) (Gornick & Sierminska 2021) and those on lower incomes fare less well in retirement preparedness than men and those on higher incomes (Noviarini et al, 2019). So what does this mean for generalizability or usefulness of your results and conclusions?
  2. What are the limitations of your study?
  3. What are your suggestions for future research?

Written expression

Generally good but a few issues that I describe below.

  1. what is “phycological” (p.1. and p. 5.) and can you list these factors, what is included in these?
  2. 1., L6 change “our” to “their”.
  3. 2., Literature Review, should “steadily decrease” be “steadily decreasing”?
  4. 3. – put SCF in full, the first time you use it
  5. 5., L2- assume the second mention of 60-40, should be 70-30
  6. 5. on H01 and H02 add “-related characteristics” at end
  7. 5 list of control variables – remaining life expectancy is listed twice
  8. 5. Eqn 7 change “rtirement” to “retirement”
  9. 5 Eqn 8 – numerator should be “retirement assets”.
  10. In the References there are a number of papers with different journal citation styles (e.g. Biggs, 2008; Smits et al, 2011; and Twenga & Campbell, 2001, 2010) that need to be changed. There are also papers that are not cited in the paper so these can be deleted as this is not a Bibliography, or alternatively cited in the paper (e.g. Butrica et al, 2009; Carter, 2018; Yaworski, 2019).

References

Gornick, J. C., & Sierminska, E. (2021). Wealth accumulation and retirement preparedness in cross-national perspective: A gendered analysis of outcomes among single adults. Journal of European Social Policy31(5), 549-564.

Goyal, K., & Kumar, S. (2021). Financial literacy: A systematic review and bibliometric analysis. International Journal of Consumer Studies45(1), 80-105.

Lührmann, M. (2010). Consumer expenditures and home production at retirement–new evidence from Germany. German Economic Review11(2), 225-245.

Noviarini, J., Coleman, A., Roberts, H., & Whiting, R. H. (2019). Housing liquidation and financial adequacy of retirees in New Zealand. Housing Studies34(9), 1543-1580.

Noviarini, J., Coleman, A., Roberts, H., & Whiting, R. H. (2021). Financial literacy, debt, risk tolerance and retirement preparedness: Evidence from New Zealand. Pacific-Basin Finance Journal68, 101598.

Wang, M., & Bartholomae, S. (2020). Retirement Preparedness: How Important Is Being Financially Literate?. Innovation in Aging4(Suppl 1), 445.

Author Response

Dear Editor,

We thank you for providing us the opportunity to revise and resubmit our manuscript. We also want to thank both the reviewers for their valuable suggestions, incorporating which have helped in substantially improving our manuscript. We have been able to address all of the reviewers’ concerns. Our point-to-point responses to the reviewer suggestions and comments are listed below.

Reviewer 2

Comments and Suggestions for Authors

Literature

  1. First line of abstract should state it is US data.

[RESPONSE] We have now made this change in our revised manuscript.

  1. The EBRI data is for what age group?

[RESPONSE] The EBRI data was for all U.S. adults 25 or older.

  1. 3. Briefly explain 401K in a footnote for non-American readers.

[RESPONSE] 401K plans are now explained in the main body of the manuscript.

  1. 5. Social security statistics from 2008 – is this still currently applicable? I would suggest that social security support has changed since 2008.

[RESPONSE] This information is directly from the Social Security Administration’s website. This number has not been updated since then. Since, this is directly from the Federal government’s website, we retained this specific reference. But in addition to this, we have also added a more recent 2022 reference of a study by the U.S. Center for Budgeting and Policy Priorities. The replacement rate still remains at approximately 40%.

  1. There is little/nothing in the way of literature to support any of your highlighted relationships in H03 between financial knowledge/literacy, education, risk tolerance and retirement preparedness (e.g. Wang & Bartholomae, 2020; Noviarini et al, 2021). This needs to be added.

[RESPONSE] We have removed the third hypothesis since this was not the focus of our study.

  1. The authors explain what is new in their study. Thank you. Can you also differentiate it from Hill (2020)?

[RESPONSE] The Hill (2020) article was a media report and not an academic article. But, we have integrate the discussion of the Hill (2020) article in the broader discussion of the rationale of our study and have also added this report in the discussions section.

  1. Although you use Life cycle theory and assume that individuals will keep constant consumption patterns, it should be acknowledged that this is not always true. There is evidence that retirees reduce consumption and expenditure due to home production, time to search for savings etc (Lührmann, 2010).

[RESPONSE] Thank you for this suggestion. We have acknowledged this as a limitation of this study.

 Method and Results

There are a number of omissions that I would expect to see:

  1. Description/Table of how variables are measured (e.g. health, risk tolerance, Financial literacy (needs explanation and reference to “the Big 3”), financial knowledge (what is the difference between financial literacy and financial knowledge?), education level

[Response] We have added Table 1 which is the description of the variables used in the study. The table shows how each variable is measured. The reference to the “Big 3” is attached as the note after Table 1. In this dataset, financial literacy is measured by the number of financial literacy questions answered correctly. We will only use financial literacy in this study to avoid confusion.

  1. What are implicates?…after much searching I found the description in https://www.federalreserve.gov/econres/scfindex.htm. This should be explained in the paper.

[RESPONSE] Following the reviewer’s suggestion, we have added an explanation of the imputation technique that was used with the five implicates in the revised version of this manuscript.

  1. Independent variables (p.5; s 4.2.8)- it is unclear what are the independent variables and what are the control variables. It appears that cohorts are the independent variables for H01 and H02 and that risk tolerance and financial literacy are the independent variables for H03. However, the authors state that the independent variables include cohort indicators, demographics, socioeconomic factors, and phycological measures and risk tolerance appears in the list of control variables.

[RESPONSE] Financial literacy and risk tolerance were controlled for in the model, but was not the focus of this analysis. Hence, we have removed H03. Also to help clear the confusion between the key independent variables of interest and other control variables, we have created two separate sub-sections when describing these variables. Additionally, table 1 has been added to explain the coding and construction of the control variables.

  1. 5. Eqn 8 What is “social replace”?

[Response] Social Replace is the social replacement ratio that is fixed at 40% in this study. Please read 4.2.6. for details.

  1. 9. Table 2, what is “Meet”?

[Response] Percentage of respondents who meet their retirement baseline.

Other issues

  1. In the calculation of dependent variable “retirement preparedness”…in Eqn6, what is the significance of “25”?; what is “R”?

[Response] “R” is the Inflation Adjusted Rate of Return. This has been described clearly in the revised manuscript.

  1. SPR – is this calculated by the authors from 1926 (pp 4-5) or did you use Damodaran (2021) (p. 5) from 1929? Also why go back such a long time, when inflation is measured from 1979 onwards?

[RESPONSE] We used data from 1928. But, this is a good point. But, given the limited availability of chronological data, the more data that is used to determine expected returns the better is the accuracy involved, hence we have kept the historic data for the stock returns. The inflation data however, was kept from 1979 because of the change in the currencies moving away from the gold standard to dollar based currency took place in the 1970’s. As a result of this structural change in the macroeconomic environment, we used inflation data from 1979

  1. 5. - how is “remaining life expectancy” measured? Is it adjusted for gender and race for each individual?

[RESPONSE] The remaining life expectancy is measured based on a response to the question in the SCF dataset, where the respondents are asked about their life expectancy. The remaining life expectancy variable is constructed by subtracting the respondents’ current ages from their self-reported life expectancy in years. A description of the construction of this variable and the Work Life Expectancy variable are added in our updated manuscript.

  1. H03- why has financial literacy just appeared? It was not mentioned in the literature section.

[RESPONSE] We have now removed this hypothesis from our revised manuscript, since financial literacy and risk tolerance were not the focus of our study.

  1. It seems that the calculations are per household (p.1.near bottom of page; p.5. just above Eqn 8) but then some data is by individual (e.g. Eqn 4, 67-age; age variable, marital status, gender, health, education). How did you deal with these different measurement units? How did you combine individual data?

[RESPONSE] the SCF responses were restricted to the primary income earners in the household. We have now listed this as a limitation of our study in our revised manuscript.

  1. You say you test for m How was this done and what are the results? For example, are age and cohorts highly correlated?

The VIF results for the variables included in our model was under 3.0, which was not a concern for multi-collinearity and was expected given the large size of this dataset (Wooldridge, 2015).

[RESPONSE] Wooldridge, J. M. (2015). Introductory econometrics: A modern approach. Cengage learning.

  1. In the testing of H03, there is the potential for endogeneity (reverse causality) (Goyal and Kumar (2021)) between financial literacy and financial outcomes (retirement preparedness). This should be considered.

[RESPONSE] We have used the financial literacy variable as a control, but we have removed H03 from our revised manuscript since this was not the focus of our study.

Conclusions

  1. There is a bias in your sample – it is only quarter female (when population % would be around 50%) and it is high income (p.4). We know that women (particularly single women) (Gornick & Sierminska 2021) and those on lower incomes fare less well in retirement preparedness than men and those on higher incomes (Noviarini et al, 2019). So what does this mean for generalizability or usefulness of your results and conclusions?

[RESPONSE] We have listed the generalizability of the findings from the empirical findings as a potential limitation of our study. This is a limitation of the SCF dataset, because the SCF oversamples higher income households. So, although these results are generalizable for the higher income Americans who are primary income earners, since majority of the primary income earners are men, this is a potential limitation of this study.

  1. What are the limitations of your study?

[RESPONSE] We have added a section on the limitations of this study.

  1. What are your suggestions for future research?

 [RESPONSE] We have added suggestions for future research in our updated manuscript.

Written expression

Generally good but a few issues that I describe below.

  1. what is “phycological” (p.1. and p. 5.) and can you list these factors, what is included in these?
  2. 1., L6 change “our” to “their”.
  3. 2., Literature Review, should “steadily decrease” be “steadily decreasing”?
  4. 3. – put SCF in full, the first time you use it
  5. 5., L2- assume the second mention of 60-40, should be 70-30
  6. 5. on H01 and H02 add “-related characteristics” at end
  7. 5 list of control variables – remaining life expectancy is listed twice
  8. 5. Eqn 7 change “rtirement” to “retirement”
  9. 5 Eqn 8 – numerator should be “retirement assets”.
  10. In the References there are a number of papers with different journal citation styles (e.g. Biggs, 2008; Smits et al, 2011; and Twenga & Campbell, 2001, 2010) that need to be changed. There are also papers that are not cited in the paper so these can be deleted as this is not a Bibliography, or alternatively cited in the paper (e.g. Butrica et al, 2009; Carter, 2018; Yaworski, 2019).

 [RESPONSE] We thank the reviewer for these valuable suggestions. We have incorporated all of the changes suggested in points 1—10 in our revised manuscript.

Round 2

Reviewer 2 Report

Thank you for addressing most of my comments and queries. Deleting H3 removed most of my theoretical and methodological concerns. I have a few remaining comments:

1.     Thank you for letting me know that EBRI data was for all US adults 25 or over. Please include this on page 3 “36% of workers aged 25 and over”

2.     Biggs and Springstead reference (page 5) has inadvertently had its date of 2008 removed. Please reinsert. It also should have the full journal title in the Reference section.

3.     Thank you for defining variables in Table 1. How do the means for Health and Education in Table 2 relate to the binary measurements specified for those variables in Table 1 (I think you actually measured them ordinally from 1-3)?

4.     The measures to do with remaining life expectancy and remaining work life expectancy are still somewhat confusing. Thank you for clarifying that life expectancy was a self- assessed measure. I think this should be mentioned in the paper (perhaps a footnote) as it is subject to a lot of inaccuracy. Then you go on to explain that remaining life expectancy (RLE) is life expectancy minus age. However, in Table 1 you have the range of RLE to be 41-150 and in Table 2 a mean of 85. This looks incorrect. This looks more like life expectancy, not RLE. I would expect perhaps 10-60 for RLE. And what about the number 150! This looks like rubbish data that should be deleted! Have you checked and cleaned the data? This makes me question whether life expectancy or RLE is used as a control in the regressions. And in Tables 5-7 you have a variable called “remaining”. I assume this is RLE, but how can I be sure? On page 6 RWLE should be defined in full words with acronym before showing its calculation.

5.     On page 2 you state that “each household will have a unique retirement baseline based on their current age, current income, expected retirement age, remaining work-life expectancy (RWLE), and remaining life expectancy (RLE)”. How does the RLE come into the calculation of the retirement baseline?

6.     The fact that the SCF responses were restricted to the primary income earners in the household should be disclosed in the data section.

7.     Can you reconcile the generational means for retirement preparedness in Table 3 and 4? They are quite different.

8.     Just before section 5 when you say that you have tested for multicollinearity, put the results (ie acceptable as VIFs<3). It provides confidence in your results.

9.     Is “income”, annual income? Someone has annual income of $676 million! Please check.

10.  Pg 6 Equation 5 – correct the spelling of “reirement”; pg 6 s.4.2.9 still have “phycological” instead of “psychological”; Pg 8 Reference to Lührmann, 2010, is not in the list of references; Pg 9 “personally” should be “personality”; Some references do not have capitals for the Journal titles

Author Response

Dear Editor,

Thank you for providing us the opportunity to revise and resubmit our manuscript. We also want to thank the reviewer for the valuable suggestions, incorporating which have helped in substantially improving our manuscript. We have been able to address all of the reviewer's concerns. Our point-to-point responses to the reviewer's suggestions and comments are listed below.

  1. Thank you for letting me know that EBRI data was for all US adults 25 or over. Please include this on page 3 “36% of workers aged 25 and over”

[Response] Adjustments have been made.

  1. Biggs and Springstead reference (page 5) has inadvertently had its date of 2008 removed. Please reinsert. It also should have the full journal title in the Reference section.

[Response] We have inserted the year and adjusted the reference. 

  1. Thank you for defining variables in Table 1. How do the means for Health and Education in Table 2 relate to the binary measurements specified for those variables in Table 1 (I think you actually measured them ordinally from 1-3)?

[Response]

We have added descriptive statistics of the binary variables to Table 2.

For Health, 1-4 corresponds to “Excellent” to “Poor.”

For Education, 1-4 corresponds to “Less than High School” to “College Degree.”

  1. The measures to do with remaining life expectancy and remaining work life expectancy are still somewhat confusing. Thank you for clarifying that life expectancy was a self- assessed measure. I think this should be mentioned in the paper (perhaps a footnote) as it is subject to a lot of inaccuracy. Then you go on to explain that remaining life expectancy (RLE) is life expectancy minus age. However, in Table 1 you have the range of RLE to be 41-150 and in Table 2 a mean of 85. This looks incorrect. This looks more like life expectancy, not RLE. I would expect perhaps 10-60 for RLE. And what about the number 150! This looks like rubbish data that should be deleted! Have you checked and cleaned the data? This makes me question whether life expectancy or RLE is used as a control in the regressions. And in Tables 5-7 you have a variable called “remaining”. I assume this is RLE, but how can I be sure? On page 6 RWLE should be defined in full words with acronym before showing its calculation.

[Response]

  • We have added a footnote about RWLE and RLE in the article (page 2).
  • We are sorry about the typo in 41-150 are the ranges of life expectancy, and thanks for noticing this mistake. We have adjusted the values of RLE and RWLE in Table 1 and 2.
  • In the regression, RLE is used as the control variable. We have changed “Remaining” to RLE to reduce confusion. RLE makes more sense than using life expectancy as the independent variable because the people should have different baselines for retirement preparedness at different life stages so the current age matters.
  • We have added the full words of RWLE on page 6.

  1. On page 2 you state that “each household will have a unique retirement baseline based on their current age, current income, expected retirement age, remaining work-life expectancy (RWLE), and remaining life expectancy (RLE)”. How does the RLE come into the calculation of the retirement baseline?

[Response] RLE should not be the element here. We have removed it from this sentence.

  1. The fact thatthe SCF responses were restricted to the primary income earners in the household should be disclosed in the data section.

[Response] We have added this fact to the data section on page 4.

  1. Can you reconcile the generational means for retirement preparedness in Table 3 and 4? They are quite different.

[Response] We have refined and rerun the codes. The adjustments have been made in Table 3 and 4, and now the results are consistent.

  1. Just before section 5 when you say that you have tested for multicollinearity, put the results (ie acceptable as VIFs<3). It provides confidence in your results.

[Response] We have made the suggested adjustments before section 5.

  1. Is “income”, annual income? Someone has annual income of $676 million! Please check.

[Response] Yes, it is annual income. The SCF oversamples high income respondents.

  1. Pg 6 Equation 5 – correct the spelling of “reirement”; pg 6 s.4.2.9 still have “phycological” instead of “psychological”; Pg 8 Reference to Lührmann, 2010, is not in the list of references; Pg 9 “personally” should be “personality”; Some references do not have capitals for the Journal titles

[Response] We have corrected all the errors mentioned above in our revised manuscript. We have checked the reference list and make sure they are in the correct format.

We noticed that several hyperlinks appeared between references on the list, and Word warns that “Error! Hyperlink reference not valid.” Did you insert these hyperlinks?

We just left those links there by assuming that you need them.
